# Registered report: BET bromodomain inhibition as a therapeutic strategy to target c-Myc

Irawati Kandela[1], Hyun Yong Jin[2], Katherine Owen[3], Reproducibility Project: Cancer Biology*

[1]Developmental Therapeutics Core, Northwestern University, Evanston, Illinois; [2]The Scripps Research Institute, La Jolla, California; [3]University of Virginia, Charlottesville, Virgina

**Abstract** The Reproducibility Project: Cancer Biology seeks to address growing concerns about reproducibility in scientific research by replicating selected results from a substantial number of high-profile papers in the field of cancer biology published between 2010 and 2012. This Registered report describes the proposed replication plan of key experiments from 'BET bromodomain inhibition as a therapeutic strategy to target c-Myc' by Delmore and colleagues, published in *Cell* in 2011 (*Delmore et al., 2011*). The key experiments that will be replicated are those reported in Figures 3B and 7C-E. Delmore and colleagues demonstrated that treatment with JQ1, a small molecular inhibitor targeting BET bromodomains, resulted in the transcriptional down-regulation of the c-Myc oncogene in vitro (Figure 3B; *Delmore et al., 2011*). To assess the therapeutic efficacy of JQ1 in vivo, mice bearing multiple myeloma (MM) lesions were treated with JQ1 before evaluation for tumor burden and overall survival. JQ1 treatment significantly reduced disease burden and increased survival time (Figure 7C-E; *Delmore et al., 2011*). The Reproducibility Project: Cancer Biology is a collaboration between the Center for Open Science and Science Exchange and the results of the replications will be published in *eLife*.

*For correspondence: tim@cos.io

Group author details
Reproducibility Project: Cancer Biology
See page 11

## Introduction

c-Myc is a DNA binding transcription factor involved in the regulation of cell proliferation, differentiation, and apoptosis (*McKeown and Bradner, 2014*). Abnormal expression of c-Myc is frequently observed in a range of malignancies including breast, colon and cervical cancer, small cell lung carcinoma, osteosarcomas, glioblastomas and myeloid leukemias (*Meyer and Penn, 2008*; *Conacci-Sorrell et al., 2014*). While c-Myc expression is required for tumor initiation and maintenance, c-Myc inactivation leads to tumor regression (*Felsher and Bishop, 1999*; *Flores et al., 2004*; *Soucek et al., 2008*; *Gabay et al., 2014*). Therefore, c-Myc represents an enticing target for pharmacological inhibition.

Multiple myeloma (MM) is an incurable disease characterized by the unrestricted proliferation of terminally differentiated plasma cells (*Anderson et al., 2011*). The primary tumor-initiating events include genetic translocation and hyperploidy, while secondary events, such as oncogenic c-Myc activation and overexpression, drive MM progression (*Bergsagel and Kuehl, 2005*; *Morgan et al., 2012*). Furthermore, studies by Chng and colleagues determined that c-Myc activation was prevalent in more than 60% of patient-derived MM cells (*Chng et al., 2011*). Despite therapeutic advances and increases in survival, patients eventually succumb to treatment-refractory disease (*Anderson, 2011*; *Kumar et al., 2012*).

Therapeutic strategies targeting c-Myc are complicated by the fact that c-Myc lacks a clear ligand-binding domain (*Darnell, 2002*). However, it is possible that c-Myc could be disrupted by other means, such as through disruption of chromatin-dependent signaling. Bromodomain and extra-terminal (BET) proteins are transcriptional regulators that epigenetically control the expression of genes involved in cell cycle, growth and inflammation (*Darnell, 2002*; *Wu and Chiang, 2007*; *LeRoy et al., 2008*; *Dey et al., 2009*; *Nicodeme et al., 2010*). BETs therefore provide potential therapeutic targets for modulating gene expression programs associated with various human diseases. Specifically, bromodomain protein 4 (BRD4), a member of the BET subfamily that associates with acetylated chromatin to promote transcription, was reported to interact with the positive transcription elongation factor complex b (P-TEFb) (*Dey et al., 2009*; *Filippakopoulos et al., 2010*). Recruitment of P-TEFb by c-Myc was also reported, providing the rationale for Delmore and colleagues to explore targeting BET proteins to inhibit c-Myc transcriptional activity (*Bisgrove et al., 2007*). Importantly, BRD4 expression was found to positively correlate with MM disease progression (*Delmore et al., 2011*). To interrogate this relationship, they used JQ1, a small molecule inhibitor of the BET family of bromodomain-containing proteins, which has the highest affinity with BRD4 and competitively inhibits BET proteins from binding to chromatin (*Filippakopoulos et al., 2010*). While the (+)-JQ1 enantiomer potently inhibits BET proteins, the (−)-JQ1 enantiomer is structurally incapable of inhibiting BET bromodomains supporting an on-target mechanism of action (*Filippakopoulos et al., 2010*). Further support for the relationship between c-Myc and BET proteins was reported by Mertz and colleagues, who used gene expression profiling of cells treated with the active and inactive forms of the JQ1 inhibitor to identify *MYC* as a highly down-regulated gene following BET bromodomain inhibition (*Mertz et al., 2011*).

As an alternative approach to direct c-Myc-targeting, Delmore and colleagues tested whether the BET inhibitor, JQ1, could effect c-Myc-specific gene silencing in MM (*Delmore et al., 2011*). In Figure 3B, Delmore and colleagues assessed the ability of JQ1 to downregulate *MYC* transcription in the MM cell line MM.1S. In this experiment, MM.1S cells were treated with JQ1 for up to 8 hours and the relative expression of *MYC* was compared to untreated control cells. JQ1 treatment resulted in a significant reduction in *MYC* transcripts as determined by qRT-PCR. This key experiment shows that JQ1 was effective at silencing *MYC* gene transcription and will be replicated in Protocol 1. Importantly, Loven and colleagues also recently corroborated these results through the demonstration that JQ1 treatment in MM.1S cells significantly decreases *MYC* mRNA levels (*Loven et al., 2013*). In addition to MM cell lines, JQ1 has proven to potently inhibit *MYC* in Merkel cell carcinoma cells (MCC-3 and 5), primary effusion lymphoma cells (PELs) and B cell acute lymphoblastic lymphomas (B-ALL) cells at the transcript level, as well as in diffuse large B cell lymphoma (DLBCL) cells at the protein expression level (*Ott et al., 2012*; *Shao et al., 2014*; *Tolani et al., 2014*; *Trabucco et al., 2015*). However, JQ1-resistant cells have also been described. Specifically, JQ1 did not alter *MYC* transcription in embryonic stem cells (ESCs) or in non-small cell lung carcinoma (NSCLC) harboring alteration in KRAS (*Shimamura et al., 2013*; *Horne et al., 2014*). In lung adenocarcinoma cells (LACs), JQ1 was found to inhibit cell growth independent of *MYC* down regulation (*Lockwood et al., 2012*).

In Figure 7C, 7D and 7E, the efficacy of JQ1 treatment was tested in mice harboring bioluminescent MM lesions. In these experiments, tumor burden was measured by whole-body bioluminescent imaging. Delmore and colleagues showed that JQ1 treatment significantly decreased disease burden and increased survival time compared to vehicle-treated control animals (*Delmore et al., 2011*). Similar findings recapitulating the suppressive effect of JQ1 on solid tumor growth have been reported in MCC, DLBCL and PEL xenograft models (*Ott et al., 2012*; *Tolani et al., 2014*; *Trabucco et al., 2015*), and reduced leukemic burden in a B-ALL xenograft model with corresponding improvements in survival (*Ott et al., 2012*). These experiments will be replicated in Protocol 2.

## Materials and methods

### Protocol 1: evaluation of *MYC* expression in JQ1-treated MM.1S cells

This experiment analyzes the expression of endogenous *MYC* during pharmacological inhibition of BET bromodomains with JQ1. This is a replication of the data presented in Figure 3B and assesses the levels of *MYC* by quantitative RT-PCR.

## Sampling

- ■ Each experiment has 9 conditions:
  - ○ qRT-PCR of *MYC* (and *GAPDH*) 0 hr after (+)-JQ1 treatment.
  - ○ qRT-PCR of *MYC* (and *GAPDH*) 1 hr after (+)-JQ1 treatment.
  - ○ qRT-PCR of *MYC* (and *GAPDH*) 8 hr after (+)-JQ1 treatment.
  - ○ qRT-PCR of *MYC* (and *GAPDH*) 0 hr after (−)-JQ1 treatment [additional].
  - ○ qRT-PCR of *MYC* (and *GAPDH*) 1 hr after (−)-JQ1 treatment [additional].
  - ○ qRT-PCR of *MYC* (and *GAPDH*) 8 hr after (−)-JQ1 treatment [additional].
  - ○ qRT-PCR of *MYC* (and *GAPDH*) 0 hr after vehicle treatment [additional].
  - ○ qRT-PCR of *MYC* (and *GAPDH*) 1 hr after vehicle treatment [additional].
  - ○ qRT-PCR of *MYC* (and *GAPDH*) 8 hr after vehicle treatment [additional].
- ■ Experiment will be performed five times with each run using three technical replicates, for a total power of ≥91%.
  - ○ See 'Power calculations' section for details.

## Materials and reagents

| Reagent | Type | Manufacturer | Catalog # | Comments |
|---|---|---|---|---|
| MM.1S-LucNeo | Cell line | Original authors | N/A | Engineered to express luciferase |
| RPMI 1640 medium | Cell culture | Sigma–Aldrich | R8758 | With 2 mM L-glutamine. Original brand not specified |
| Fetal bovine serum (FBS) | Cell culture | Sigma–Aldrich | F0392 | Original brand not specified |
| 100× Penicillin/streptomycin | Cell culture | Sigma–Aldrich | P4333 | Original brand not specified |
| PBS, without $MgCl_2$ and $CaCl_2$ | Buffer | Sigma–Aldrich | D8537 | Originally not specified |
| 0.05% trypsin/0.48 mM EDTA | Cell culture | Sigma–Aldrich | T3924 | Originally not specified |
| 35-mm tissue culture dishes | Labware | Corning | 430165 | Originally not specified |
| (+)-JQ1 enantiomer | Chemical | EMD Millipore | 500586 | Original made by authors |
| (−)-JQ1 enantiomer | Chemical | | | |
| DMSO | Chemical | Sigma–Aldrich | D8418 | Original brand not specified |
| TRI reagent | Chemical | Sigma–Aldrich | T9424 | Replaces TRIzol from Invitrogen (Cat #15596-026) |
| First-Strand cDNA Synthesis kit | Nucleic acid | GE Healthcare (Sigma–Aldrich) | GE27-9261-01 | – |
| Real-time PCR system | Instrument | Applied Biosystems | 7900HT | Replaces 7500 model |
| TaqMan Gene Expression Master Mix | Nucleic acid | Life Technologies | 4369016 | Replaces a real-time PCR kit from Applied Biosystems (Cat #N15597), which is discontinued |
| Taq-Man probe (*MYC*) | Nucleic acid | Applied Biosystems | Hs00905030_m1 | – |
| Taq-Man probe (*Gapdh*) | Nucleic acid | Applied Biosystems | Hs02758991_g1 | – |

## Procedure

### Notes

- All cells will be sent for mycoplasma testing and STR profiling.
- Cells maintained in RPMI 1640 with 2 mM L-glutamine supplemented with 10% FBS, 100 U/ml penicillin, and 50 μg/ml streptomycin at 37°C in a humidified atmosphere at 5% $CO_2$.

1. Seed $8 \times 10^5$ MM.1S-LucNeo cells into three 35-mm tissue culture dishes.
2. The next day treat the dishes of cells with 2 ml of media with a final concentration of 500 nM (+)-JQ1, 500 nM (−)-JQ1, or an equivalent volume of DMSO.
   a. Make 10 mM stock of (+)-JQ1 and (−)-JQ1 by diluting in DMSO.
3. Isolate RNA from dishes at the following time points after treatment using TRI Reagent following manufacturer's instructions.
   a. 0 hr (immediately).
   b. 1 hr.
   c. 8 hr.

4. Reverse transcribe total RNA to cDNA with reverse transcription kit following manufacturer's instructions.
   a. Record RNA concentration and purity.
   b. Use 1 µg of RNA per 50 µl reaction.
   c. Use random hexamers for first-strand synthesis.
5. Perform qPCR to assess *MYC* expression levels using a real-time PCR system with a real-time PCR kit following manufacturer's instructions. Perform triplicate technical replicates for each biological replicate.
   a. Use 5 µl of undiluted cDNA mixture per 50 µl reaction.
   b. Use TaqMan probes for *MYC* (Hs00905030_m1) and *Gapdh* (Hs02758991_g1).
6. Analyze and compute $\Delta\Delta C_T$ values.
   a. The first qRT-PCR assay will be analyzed to ensure conditions are appropriate for proper quantitation. If it is determined that conditions need to be adjusted, such as input volume, the conditions will be adjusted and the reaction will be repeated. Once optimized, the conditions will be used for all subsequent reactions.
      i. All details and data associated with this process will be recorded.
7. Repeat steps 1–6 independently four additional times.

## Deliverables

- Data to be collected:
  - Purity ($A_{260/280}$ ratio) and concentration of isolated total RNA from cells.
  - Assay conditions used initially and, if necessary, modified, to ensure conditions are appropriate for proper quantitation.
  - Raw qRT-PCR values, as well as analyzed $\Delta\Delta C_T$ values.
  - Bar graph of *MYC* mRNA levels normalized to 0 hr after (+)-JQ1 treatment. (Compare to Figure 3B).

## Confirmatory analysis plan

This replication attempt will perform the following statistical analysis listed below.

- Statistical analysis:
  - Repeated measures ANOVA of normalized *MYC* mRNA levels in MM.1S cells treated with (+)-JQ1, (−)-JQ1, or DMSO.
    - Paired *t*-tests with the Bonferroni correction:
      1. MM.1S cells harvested 8 hr after (+)-JQ1 treatment compared to cells 0 hr after (+)-JQ1 treatment.
      2. MM.1S cells harvested 1 hr after (+)-JQ1 treatment compared to cells 0 hr after (+)-JQ1 treatment.
- Additional exploratory statistical Analysis:
  - Two-way ANOVA of normalized *MYC* mRNA levels in MM.1S cells treated with (+)-JQ1, (−)-JQ1, or DMSO.
    - Planned comparisons with the Bonferroni correction:
      1. MM.1S cells harvested 8 hr after (+)-JQ1 treatment compared to cells 0 hr after (+)-JQ1 treatment.
      2. MM.1S cells harvested 1 hr after (+)-JQ1 treatment compared to cells 0 hr after (+)-JQ1 treatment.
- Meta-analysis of effect sizes:
  - Compute the effect sizes of each comparison, compare them against the effect size in the original paper and use a random effects meta-analytic approach to combine the original and replication effects, which will be presented as a forest plot.

## Known differences from the original study

The replication experiment will not include the full time course treatment with (+)-JQ1, but will only include the 0 hr, 1 hr, and 8 hr treatments. The replication experiment will also include six additional conditions; MM.1S cells harvested 0 hr, 1 hr, and 8 hr after treatment with (−)-JQ1 or DMSO. The original report was unclear if the cells used for this experiment were MM.1S or MM.1S-LucNeo cells. This replication attempt will use MM.1S-LucNeo cells since this is the same cell line used in protocol 2 to assess the efficacy of JQ1 treatment in mice. The original details for the reverse transcription and qRT-PCR reactions were not known, thus the details used here are manufacturer recommended with the use of random hexamers to generate cDNA that are more representative of all regions of the transcripts. Following the first reaction, conditions will be adjusted if necessary to ensure proper quantitation. All known differences of materials and reagents are listed in the 'Materials and reagents' section above with the originally used item listed in the comments section. All differences have the same capabilities as the original and are not expected to alter the experimental design.

## Provisions for quality control

The cell lines used in this experiment will undergo STR profiling to confirm their identity and will be sent for mycoplasma testing to ensure there is no contamination. The sample purity ($A_{260/280}$ and $A_{260/230}$ ratios) of the isolated RNA from each sample will be reported. The first qRT-PCR reaction will be analyzed to determine if the assay conditions are appropriate for proper quantitation. This would be apparent after the first qRT-PCR assay, and the assay would then be repeated adjusting parameters. After conditions that allow for proper quantitation are achieved, then the number of pre-determined replicates will be completed under those same conditions. All assay details, adjustments, and data will be recorded and made available during this optimization process. All data obtained from the experiment—raw data, data analysis, control data, and quality control data—will be made publicly available, either in the published manuscript or as an open access data set available on the Open Science Framework project page for this study (https://osf.io/7zqxp/).

## Protocol 2: JQ1 treatment in orthotopic xenograft model

This experiment tests the efficacy of JQ1 treatment in mice harboring bioluminescent MM lesions. This is a replication of the data presented in Figures 7C, 7D, and 7E and assesses tumor burden by whole-body bioluminescent imaging and monitors overall survival with daily treatment of JQ1.

### Sampling

- Each experiment has 3 cohorts:
  - Cohort 1: Vehicle (D5W) treatment.
  - Cohort 2: (+)-JQ1 treatment.
  - Cohort 3: (−)-JQ1 treatment [additional].
- Experiment will analyze at least eight mice per cohort for a minimum of 80% power.
  - See 'Power calculations' section for details.
- To account for unexpected euthanasia of mice before the end of the experiment or exclusion of mice before treatment, the sample size was increased by ~ 30%.
  - A total of 33 mice will be injected with MM.1S-LucNeo cells.

### Materials and reagents

| Reagent | Type | Manufacturer | Catalog # | Comments |
|---|---|---|---|---|
| MM.1S-LucNeo | Cell line | Original authors | N/A | Engineered to express luciferase |
| RPMI 1640 medium | Cell culture | Sigma–Aldrich | R8758 | With 2 mM L-glutamine. Original brand not specified |
| Fetal bovine serum (FBS) | Cell culture | Sigma–Aldrich | F0392 | Original brand not specified |
| 100× penicillin/streptomycin | Cell culture | Sigma–Aldrich | P4333 | Original brand not specified |
| PBS, without $MgCl_2$ and $CaCl_2$ | Buffer | Sigma–Aldrich | D8537 | Originally not specified |
| 0.05% trypsin/0.48 mM EDTA | Cell culture | Sigma–Aldrich | T3924 | Originally not specified |
| T150 tissue culture flasks | Labware | Corning | 430825 | Originally not specified |
| 5 week old female Fox Chase SCID Beige (CB17.Cg-$Prkdc^{scid}Lyst^{bg-J}$/Crl) | Animal model | Charles River Labs | Strain 250 | – |
| 30½G needle | Labware | Sigma–Aldrich | Z192341 | Originally not specified |
| 27½G needle | Labware | Sigma–Aldrich | Z192384 | Originally not specified |
| 1 ml syringe | Labware | Sigma–Aldrich | Z192090 | Originally not specified |
| VivoGlo luciferin | Reporter assay | Promega | P1042 | Original catalog # not specified |
| Xenogen IVIS Spectrum | Instrument | Caliper Life Sciences | Spectrum | – |
| Living Images | Software | Caliper Life Sciences | Version used will be recorded and included in the Replication Study | |
| (+)-JQ1 enantiomer | Chemical | EMD Millipore | 500586 | Original made by authors |
| (−)-JQ1 enantiomer | Chemical | | | |
| Dextrose (D-(+)-glucose) | Chemical | Sigma–Aldrich | G8270 | Original brand not specified |

## Procedure

### Notes

- All cells will be sent for mycoplasma testing and STR profiling, as well as screened against a Rodent Pathogen Panel.
- Cells maintained in RPMI 1640 with 2 mM L-glutamine supplemented with 10% FBS, 100 U/ml penicillin, and 50 μg/ml streptomycin at 37°C in a humidified atmosphere at 5% $CO_2$.
- MM.1S-LucNeo cells stably express a luciferase construct.

1. After 1 week of acclimation, intravenously inject $2 \times 10^6$ MM.1S-LucNeo cells suspended in 200 μl PBS into 6-week old female SCID-beige mice using a 30½G needle via the lateral tail vein.
2. 8 days later, inject mice intraperitoneally with 75 mg/kg of D-luciferin in 0.1 ml using a 27½G needle. Anesthetize mice and image using a Xenogen IVIS Spectrum using the Living Images software package.
   a. Record weight of mice.
   b. Anesthetize with isoflurane.
   c. Image mice 20 min post injection.
3. 5 days later, inject mice with 75 mg/kg of D-luciferin as described in step 2 and image using a Xenogen IVIS Spectrum using the Living Images software package.
   a. Record weight of mice.
   b. Determine difference between bioluminescence of first imaging and second imaging.
4. For mice with established disease, randomly divide into three cohorts.
   a. Established disease is defined as detection of MM.1S-LucNeo lesions diffusely engrafted in the skeleton with an increase in bioluminescence between the first and second images.
   b. Exclude any mice with no detectable disease or no increase in bioluminescence. If over 30 mice are present with detectable disease, exclude mice with lowest disease burden to obtain 30 mice for randomization and treatment.
      i. Original report saw 90–95% engraftment.
   c. Animals are ranked according to disease burden (difference between bioluminescence of first imaging and second imaging), to balance groups for baseline tumor characteristics, and assigned to group 1, group 2, or group 3 using an alternating serpentine method.
      (rank 1 = group 1, rank 2 = group 2, rank 3 = group 3, rank 4 = group 3, rank 5 = group 2, rank 6 = group 1, rank 7 = group 1, etc).
      i. Designation of Vehicle, (+)-JQ1, or (−)-JQ1 treatments as group 1, group 2, or group 3 will be determined by randomly assigning the three treatments into one block using www.randomization.com.
      ii. Record seed number.
5. After imaging and randomization, treat mice daily with either (+)-JQ1 at 50 mg/kg, (−)-JQ1 at 50 mg/kg, or vehicle (5% dextrose in water) control by intraperitoneal injection with a 27½G needle.
   a. Inject 10 ml/kg body weight of a 5 mg/ml solution to give a final dose of 50 mg/kg.
      i. (+)-JQ1 and (−)-JQ1 solutions are prepared in 5% dextrose in water.
   b. Record weight of mice.
6. 6, 14, and 21 days after the start of treatment IP injections, assess tumor burden by bioluminescence imaging after IP injection of 75 mg/kg of D-luciferin as described in step 2 above.
   a. Record weight of mice.
7. Continue to treat mice with daily injections of (+)-JQ1, (−)-JQ1, or vehicle control until mice are euthanized according to IACUC guidelines or until end of experiment (5 weeks total treatment).
   a. In this model, mice are euthanized when they develop hind limb paralysis.

### Deliverables

- Data to be collected:
  - Mouse health records (including number of mice with established disease and reason for euthanasia, weight at time of each injection).
  - All images of mice in vivo to detect established disease and tumor burden (compare to Figure 7C).
  - Raw photon flux measurements of each mouse and graph of bioluminescence vs day of treatment of cohorts (compare to Figure 7D).
  - Raw survival data and Kaplan–Meier curves generated for percent survival (compare to Figure 7E).

## Confirmatory analysis plan

This replication attempt will perform the following statistical analyses listed below.

- Statistical analysis:
  - Tumor burden:
    - One-way ANOVA test on day 22 data points with the following planned comparisons using Fisher's LSD correction:
      i. (+)-JQ1 treatment to vehicle treatment.
      ii. (+)-JQ1 treatment to (−)-JQ1 treatment.
    - One-way ANCOVA test of the area under the curve (AUC) measurements (determined from day 1, 7, 15, and 22 data for each mouse) with the AUC pre-treatment measurements (determined from day −4 and 1 for each mouse) as the covariate, with the following planned comparisons with the Bonferroni correction:
      i. (+)-JQ1 treatment to vehicle treatment.
      ii. (+)-JQ1 treatment to (−)-JQ1 treatment.
      Note: This is an additional test not originally performed, which analyzes all data points opposed to just day 22 data points.
  - Kaplan–Meier curves:
    - Log-rank Mantel–Cox test on the following comparisons with the Bonferroni correction:
      i. (+)-JQ1 treatment to vehicle treatment.
      ii. (+)-JQ1 treatment to (−)-JQ1 treatment.
- Meta-analysis of effect sizes:
  - Compute the effect sizes of each comparison, compare them against the effect size in the original paper and use a random effects meta-analytic approach to combine the original and replication effects, which will be presented as a forest plot.

## Known differences from the original study

The replication experiment will include an additional cohort receiving treatment with the inactive (−)-JQ1 enantiomer, which was not included in the original study. All known differences of materials and reagents are listed in the 'Materials and reagents' section above with the originally used item listed in the comments section. All differences have the same capabilities as the original and are not expected to alter the experimental design.

## Provisions for quality control

The cell lines used in this experiment will undergo STR profiling to confirm their identity and will be sent for mycoplasma testing to ensure there is no contamination. Additionally, cells used for xenograft injection will be screened against a Rodent Pathogen Panel to ensure no contamination prior to injection. The bioluminescence images and measurements will be reported for all mice when determining the inclusion based on detection of lesions diffusely engrafted in the skeleton. Mice will be randomly assigned to treatment group with disease burden balanced among groups with the seed number recorded to reproduce the plan. All data obtained from the experiment—raw data, data analysis, control data, and quality control data—will be made publicly available, either in the published manuscript or as an open access data set available on the Open Science Framework project page for this study (https://osf.io/7zqxp/).

## Power calculations

For additional details on power calculations, please see analysis scripts and associated files on the Open Science Framework:

- https://osf.io/bjrpc/.

### Protocol 1

Summary of original data presented in Figure 3B (estimated from graph).

| Dataset being analyzed | N | Mean | SD |
|---|---|---|---|
| MM.1S cells treated with (+)-JQ1–0 hr | 2* | 1.0 | 0.375 |
| MM.1S cells treated with (+)-JQ1–1 hr | 2* | 0.06875 | 0.00625 |
| MM.1S cells treated with (+)-JQ1–8 hr | 2* | 0.0875 | 0.05 |

*This is the number of biological replicates reported for this experiment.

We are including the following groups in the replication study: (−)-JQ1 and vehicle treatment. We performed these calculations with the assumption that (−)-JQ1–0 hr, 1 hr, and 8 hr, and vehicle–0 hr, 1 hr, and 8 hr will have similar values as (+)-JQ1–0 hr.

## Analysis with samples paired

### Test family

- ■ ANOVA: Repeated measures, within factors, alpha error = 0.05.
  - Power calculations performed with G*Power software, version 3.1.7 (*Faul et al., 2007*).
    - ○ The correlation among repeated measures was assumed to be 0 and the nonsphericity correction was assumed to be 1.

| Time | Groups | Detectable effect size $f$ | A priori power | Total sample size |
|---|---|---|---|---|
| 0 hr, 1 hr, and 8 hr | (+)-JQ1, (−)-JQ1*, vehicle* | 0.41611† | 80.0%† | 15‡ (3 groups, 3 measurments) |

*(−)-JQ1 and vehicle values were the same as (+)-JQ1–0 hr values for this calculation.
†This is the effect size detectable with 80% power and the indicated sample size.
‡A total sample size of 15 will be used based on the paired $t$-test planned comparison calculations.

### Test family

- 2 tailed $t$ test, difference between two dependent means (matched pairs): Bonferroni's correction: alpha error = 0.025.
  - ■ Power calculations performed with G*Power software, version 3.1.7 (*Faul et al., 2007*).
    - ○ Correlation between groups was assumed to be 0.

| Group 1 | Group 2 | Effect size $d_z$ | A priori power | Total sample size |
|---|---|---|---|---|
| (+)-JQ1–0 hr | (+)-JQ1–8 hr | 2.41199 | 90.8% | 5 |
| (+)-JQ1–0 hr | (+)-JQ1–1 hr | 2.48299 | 92.3% | 5 |

## Analysis with samples unpaired

### Test family

- ■ Two-way ANOVA: Fixed effects, special, main effects and interactions, alpha error = 0.05.
  - Power calculations performed with G*Power software, version 3.1.7 (*Faul et al., 2007*).
  - ANOVA F statistic calculated with R software 3.1.2 (*Team RC, 2014*).
  - F test statistic (interaction) calculated from *Cohen (2002)*.
  - Partial $\eta^2$ calculated from *Lakens (2013)*.

| Time | Groups | F test statistic | Partial $\eta^2$ | Effect size $f$ | A priori power | Total sample size |
|---|---|---|---|---|---|---|
| 0 hr, 1 hr, and 8 hr | (+)-JQ1, (−)-JQ1*, vehicle* | F(4,9) = 1.7228 (interaction) | 0.4336 | 0.87503 | 82.4%† | 23† (9 groups) |

*(−)-JQ1 and vehicle values were the same as (+)-JQ1–0 hr values for this calculation.
†A total sample size of 45 will be used based on the paired $t$-test planned comparison calculations making the power 99.7%.

### Test family

- 2 tailed $t$ test, difference between two independent means: Bonferroni's correction: alpha error = 0.025.

Power calculations performed with G*Power software, version 3.1.7 (*Faul et al., 2007*).

| Group 1 | Group 2 | Effect size *d* | A priori power | Group 1 sample size | Group 2 sample size |
|---|---|---|---|---|---|
| (+)-JQ1–0 hr | (+)-JQ1–8 hr | 3.41107 | 93.3%* | 4* | 4* |
| (+)-JQ1–0 hr | (+)-JQ1–1 hr | 3.51148 | 95.5%† | 4† | 4† |

*5 per group will be used based on the paired *t*-test planned comparisons making the power 98.7%.
†5 per group will be used based on the paired *t*-test planned comparisons making the power 99.1%.

## Protocol 2
## Tumor burden as determined by bioluminescence
Summary of original data presented in Figure 7D (provided by authors).

| Dataset being analyzed | Day | N | Mean | SD |
|---|---|---|---|---|
| Vehicle-treated mice | −4 | 10 | $2.14 \times 10^6$ | $9.38 \times 10^5$ |
| | 1 | 10 | $1.06 \times 10^7$ | $7.63 \times 10^6$ |
| | 7 | 10 | $2.37 \times 10^8$ | $7.98 \times 10^7$ |
| | 15 | 10 | $6.40 \times 10^9$ | $3.13 \times 10^9$ |
| | 22 | 10 | $1.85 \times 10^{10}$ | $1.01 \times 10^{10}$ |
| (+)-JQ1-treated mice | −4 | 9 | $1.59 \times 10^6$ | $2.84 \times 10^5$ |
| | 1 | 9 | $8.10 \times 10^6$ | $3.31 \times 10^6$ |
| | 7 | 9 | $5.78 \times 10^7$ | $2.84 \times 10^7$ |
| | 15 | 9 | $1.10 \times 10^9$ | $5.70 \times 10^8$ |
| | 22 | 9 | $5.52 \times 10^9$ | $2.25 \times 10^9$ |

Area under the curve (AUC) calculations from estimated values from graph in Figure 7D. Calculations performed with R software 3.1.2 (*Team RC, 2014*).

| Data set being analyzed | Days | N | Mean | SD |
|---|---|---|---|---|
| Vehicle-treated mice | −4 to 1 | 10 | $3.02 \times 10^7$ | $2.30 \times 10^7$ |
| | 1 to 22 | 10 | $1.14 \times 10^{11}$ | $5.08 \times 10^{10}$ |
| (+)-JQ1-treated mice | −4 to 1 | 10 | $2.51 \times 10^7$ | $9.46 \times 10^6$ |
| | 1 to 22 | 10 | $2.80 \times 10^{10}$ | $8.72 \times 10^9$ |

We are including the following group in the replication study: (−)-JQ1-treated mice. We performed these calculations with the assumption that (−)-JQ1-treated mice will have similar values as vehicle-treated mice.

## Day 22 values

### Test family

- F test: ANOVA: Fixed effects, omnibus, one-way, alpha error = 0.05.
  - Power calculations performed with G*Power software, version 3.1.7 (*Faul et al., 2007*).
  - ANOVA F test statistic calculated with R software 3.1.2 (*Team RC, 2014*).
  - F test statistic (interaction) calculated from *Cohen (2002)*.
  - Partial $\eta^2$ calculated from *Lakens (2013)*.

| Groups | Data | F test statistic | Partial η² | Effect size f | A priori power | Total sample size |
|---|---|---|---|---|---|---|
| (+)-JQ1, (−)-JQ1*, vehicle | 22 days | F(2,26) = 7.2114 | 0.35680 | 0.74480 | 80.8%† | 21† (3 groups) |

*(−)-JQ1 values were the same as vehicle values for this calculation.
†8 per group (24 total) will be used based on the planned comparisons making the power 86.8%.

## Test family

- 2 tailed *t* test: Means: Difference between two independent means: Fisher's LSD correction: alpha error = 0.05.
  - Power calculations performed with G*Power software, version 3.1.7 (*Faul et al., 2007*).

| Group 1 | Group 2 | Effect size d | A priori power | Group 1 sample size | Group 2 sample size |
|---|---|---|---|---|---|
| (+)-JQ1 | Vehicle | 1.52437 | 80.9% | 8 | 8 |
| (+)-JQ1 | (−)-JQ1 | 1.52437 | 80.9% | 8 | 8 |

## AUC values

### Test family

- F test: ANCOVA: Fixed effects, main effects and interactions, alpha error = 0.05.
  - Power calculations performed with G*Power software, version 3.1.7 (*Faul et al., 2007*).
  - ANCOVA F test statistic calculated with R software 3.1.2 (*Team RC, 2014*).
  - Partial η² calculated from *Lakens (2013)*.

| Groups | Data | F test statistic | Partial η² | Effect size f | A priori power | Total sample size |
|---|---|---|---|---|---|---|
| (+)-JQ1, (−)-JQ1*, vehicle | AUC† | F(2,25) = 36.051 | 0.74254 | 1.69826 | 98.3%‡ | 11‡ (3 groups) |

*(−)-JQ1 values were the same as vehicle values for this calculation.
†One covariate was used (days −4 to 1 AUC) for this calculation.
‡8 per group (24 total) will be used based on the day 22 calculations making the power 99.9%.

## Test family

- 2 tailed *t* test: Means: Difference between two independent means: Bonferroni correction: alpha error = 0.025.
  - Power calculations performed with G*Power software, version 3.1.7 (*Faul et al., 2007*).

| Group 1 | Group 1 adjusted mean | Group 2 | Group 2 adjusted mean | Effect size d | A priori power | Group 1 sample size | Group 2 sample size |
|---|---|---|---|---|---|---|---|
| (+)-JQ1 | $1.12 \times 10^{11}$ | Vehicle | $3.45 \times 10^{10}$ | 3.43522 | 93.6%* | 4* | 4* |
| (+)-JQ1 | $1.12 \times 10^{11}$ | (−)-JQ1 | $3.45 \times 10^{10}$ | 3.43522 | 93.6%* | 4* | 4* |

*8 per group will be used based on the day 22 calculations making the power 99.9%.

## Survival data
Summary of original data presented in Figure 7E (provided by authors).

| Dataset being analyzed | Median survival | Hazard ratio (to vehicle control) | N |
|---|---|---|---|
| Vehicle-treated mice | 22 days | NA | 10 |
| JQ1-treated mice | 35 days | 0.038565 | 9* |

*Only 9 animals were analyzed in this group.

We are including the following comparisons in the replication study: (+)-JQ1-treated mice to (−)-JQ1-treated mice. We performed these calculations with the assumption that (−)-JQ1-treated mice will have similar values as vehicle-treated mice.

## Test family

- Log-rank (Mantel–Cox) test: Bonferroni correction: alpha error = 0.025.
  - Power calculations performed with the Sample Size Calculator (*Schoenfeld, 1983*).

| Group 1 | Group 2 | Experiment duration | A priori power | Total events needed | Group 1 sample size | Group 2 sample size |
|---------|---------|---------------------|----------------|---------------------|---------------------|---------------------|
| (+)-JQ1 | Vehicle | 40 days | 80% | 4* | 5* | 5* |
| (+)-JQ1 | (−)-JQ1 | 40 days | 80% | 4* | 5* | 5* |

*7 per group will be used based on the bioluminescence analysis making the power 94%.

## Acknowledgements

The Reproducibility Project: Cancer Biology core team would like to thank the original authors, in particular Dr Andrew Kung, for generously sharing critical information as well as reagents to ensure the fidelity and quality of this replication attempt. We thank Courtney Soderberg at the Center for Open Science for assistance with statistical analyses. We would also like to thank the following companies for generously donating reagents to the Reproducibility Project: Cancer Biology; American Tissue Culture Collection (ATCC), BioLegend, Charles River Laboratories, Corning Incorporated, DDC Medical, EMD Millipore, Harlan Laboratories, LI-COR Biosciences, Mirus Bio, Novus Biologicals, Sigma–Aldrich and System Biosciences (SBI).

## Additional information

### Group author details

**Reproducibility Project: Cancer Biology**

Elizabeth Iorns: Science Exchange, Palo Alto, California; William Gunn: Mendeley, London, United Kingdom; Fraser Tan: Science Exchange, Palo Alto, California; Joelle Lomax: Science Exchange, Palo Alto, California; Nicole Perfito: Science Exchange, Palo Alto, California; Timothy Errington: Center for Open Science, Charlottesville, Virginia

### Competing interests

IK: This is a Science Exchange associated lab. RP:CB: EI, FT, JL, and NP: Employed by and hold shares in Science Exchange Inc. The other authors declare that no competing interests exist.

### Funding

| Funder | Author |
|--------|--------|
| Laura and John Arnold Foundation | Reproducibility Project: Cancer Biology |

The Reproducibility Project: Cancer Biology is funded by the Laura and John Arnold Foundation, provided to the Center for Open Science in collaboration with Science Exchange. The funder had no role in study design or the decision to submit the work for publication.

### Author contributions

IK, HYJ, KO, Drafting or revising the article; RP:CB, Conception and design, Drafting or revising the article

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
