## [Decision Letter]

Thank you for sending your work entitled “Registered report: BET bromodomain inhibition as a therapeutic strategy to target c-Myc” for consideration at *eLife*. Your article has been favorably evaluated by Charles Sawyers (Senior editor), a Reviewing editor, and three reviewers.

The Reviewing editor and the reviewers discussed their comments before we reached this decision, and the Reviewing editor has assembled the following comments to help you prepare a revised submission.

This Registered Report submission proposes to replicate key findings in [9], which reported the ability of the BRD inhibitor JQ1 to suppress *MYC* expression in a myeloma cell line and to extend survival of mice transplanted with this cell line. There was general agreement among the three Referees that the proposed studies will assess the major findings of Delmore et al., and will be highly relevant. The referees made the following recommendations to improve the protocols.

Protocol 1:

One reviewer noted that one of the main strengths of the published study (9) was that JQ1 treatment led to downregulation of Myc was not limited to MM1.S cells, and that the effect was observed in several MM cell lines (Figure 3H and 3I). This is a key observation and we recommend that the qPCR analyses be extended to the MM cell types indicated in Figures 2A and 3I.

Addition of one more time point is recommended, the 1h treatment with 500 nM (+)-JQ1. This will indicate whether the Myc downregulation by JQ1 is as dynamic as reported.

In addition to the qPCR primers for Myc and GAPDH mentioned, the exact qPCR primers used by (9) should be included.

One reviewer requested scripts and a detailed description of the calculation performed with R. For example in Protocol 1, in the subsection headed “Test family”, the following sentence can be added: F test statistic (interaction) has been calculated following [6] and the partial η2 has been calculated following [18].

This reviewer disagrees with the choice of a simple two-way ANOVA. In the original paper (Figure 3B) paired Student's *t*-tests were used. Since the replicated experiments are similar to the original ones a repeated measures anova is more appropriate. A carefully chosen repeated measures ANOVA is a natural extension of the paired *t*-test and can simplify the implementation of the proposed meta-analysis. A drawback with a repeated measures ANOVA is that it can be more difficult to set the parameters to determine the power. In such a case a sensitivity analysis can be performed with the G* power software.

Protocol 2:

Weight of mice should be recorded at day-0/day of injection.

In this protocol the analyses following an ANOVA have been performed using Fisher's LSD correction and alpha error = 0.05. In its basic form (I think the one used in the protocol) the LSD correction is not taking into account that multiple comparisons will be performed and therefore a Bonferroni correction (or other corrections) must be employed. For Protocol 2 this brings alpha to 0.025 and in practice is not dramatically changing the power calculations. As an alternative to Fisher's LSD followed by Bonferroni, the Hayter-Fisher's LSD procedure (Hayter, 1986) controls the MFWER (maximum family wise error rate).

Fisher's LSD correction has been reported also for survival data but it doesn't apply to this kind of data. Also here we need a (Bonferroni) correction. For the survival data, power calculations were performed with the Sample Size Calculator, however I do not have a clear link to the software. The authors should provide all the used parameters and references.

References:

Anthony J. Hayter. The maximum familywise error rate of fisher's least significant difference test. Journal of the American Statistical Association, 81(396):1000–1004, 1986. doi:10.1080/01621459.1986.10478364

---

## [Author Response]

*Protocol 1*:

*One reviewer noted that one of the main strengths of the published study (*[9]*) was that JQ1 treatment led to downregulation of Myc was not limited to MM1.S cells, and that the effect was observed in several MM cell lines (Figure 3H and 3I). This is a key observation and we recommend that the qPCR analyses be extended to the MM cell types indicated in Figures 2A and 3I*.

We agree that testing additional cell types (KMS11, OPM1, LR5, and INA6) provided additional evidence that the downregulation of Myc was not limited to MM1.S cells, however the Reproducibility Project: *Cancer* Biology aims to perform direct replications using the same methodology reported in the original paper. The additional cell types would be a conceptual replication, which we agree is a useful approach to test the experiment’s underlying hypothesis, but which is not an aim of the project. Aspects of an experiment not included in the original study are occasionally added to ensure the quality of the research, but by no means is a requirement of this project; rather, it is an extension of the original work. Adding additional aspects not included in the original study can be of scientific interest, and can be included if it is possible to balance them with the main aim of this project: to perform a direct replication of the original experiment(s). As such, we will restrict our analysis to the experiments being replicated and will not include discussion of experiments not being replicated in this study.

*Addition of one more time point is recommended, the 1h treatment with 500 nM (+)-JQ1. This will indicate whether the Myc downregulation by JQ1 is as dynamic as reported*.

Thank you for the recommendation. We have updated the manuscript to reflect this additional time point.

*In addition to the qPCR primers for Myc and GAPDH mentioned, the exact qPCR primers used by (*[9]*) should be included*.

The qPCR primers included in this experimental design were reported in [9] (Supplemental information, Extended Materials and methods, Expression analysis).

*One reviewer requested scripts and a detailed description of the calculation performed with R. For example in Protocol 1, in the subsection headed “Test family”, the following sentence* can *be added: F test statistic (interaction) has been calculated following*
[6]
*and the partial η2 has been calculated following*
[18].

Thank you for this recommendation. We have included a link to the scripts (https://osf.io/bjrpc/?view_only=737ba0f51c474aa1bc2782a44fba34d5). Additionally, we have added descriptions as suggested to more clearly describe the approach.

*This reviewer disagrees with the choice of a simple two-way ANOVA. In the original paper (Figure 3B) paired Student's* t*-tests were used. Since the replicated experiments are similar to the original ones a repeated measures ANOVA is more appropriate. A carefully chosen repeated measures ANOVA is a natural extension of the paired* t*-test and* can *simplify the implementation of the proposed meta-analysis. A drawback with a repeated measures ANOVA is that it* can *be more difficult to set the parameters to determine the power. In such a case a sensitivity analysis* can *be performed with the G* power software*.

We thank the reviewer for catching the original analysis method. We have adjusted the planned analysis to include this approach. As the reviewer suggested, we conducted a sensitivity analysis for the repeated measures ANOVA using the planned sample size and assuming 0 correlation among repeat measures and a nonsphericity correction of 1 to allow for a conservative estimate. We included the two planned comparisons (paired *t*-tests) assuming a correlation between groups of 0. This is because we do not have access to the original raw data. Additionally, it is not clear what df was used in the original paper since the reported p values suggest a larger effect size estimate than using the estimated means and standard deviations reported in the figure. This is likely due to the combination of the technical replicates (3) being combined with the biological replicates (2) giving a df of 5, opposed to using only the biological replicates (what is proposed in this manuscript), which gives a df of 2. Thus, we used the point estimated from G*Power using the estimated values from the graph reported in Figure 3B.

We are also proposing to analyze the data as originally planned (as a between subjects design) since the experimental set-up suggests this analysis. The sample comes from multiple random dishes of cells treated with or without drug. As such, it is possible to have a different number of data points in one group (vehicle) than the other (JQ1), thus making matched samples difficult. We think it is reasonable to use the independent test, and both analysis designs are reported in the literature. This will be considered additional exploratory analysis since it was not originally reported.

*Protocol 2*:

*Weight of mice should be recorded at day-0/day of injection*.

This is a parameter we have in the manuscript. Protocol 2, Step 5b. Weight of mice will be recorded during each day of injection during the course of the experiment.

*In this protocol the analyses following an ANOVA have been performed using Fisher's LSD correction and alpha error = 0.05. In its basic form (I think the one used in the protocol) the LSD correction is not taking into account that multiple comparisons will be performed and therefore a Bonferroni correction (or other corrections) must be employed. For Protocol 2 this brings alpha to 0.025 and in practice is not dramatically changing the power calculations. As an alternative to Fisher's LSD followed by Bonferroni, the Hayter-Fisher's LSD procedure (Hayter, 1986) controls the MFWER (maximum family wise error rate)*.

We agree with the reviewers comment on the use of a correction, such as Bonferroni or the modification of LSD by Hayter are ways to control for the MFWER, however as Hayter describes in his 1986 paper, this applies in situations where the ANOVA is unbalanced or with a balanced design with four or more populations. Since the proposed analysis is balanced with three population groups, the LSD is sufficiently conservative and powerful to account for the multiple comparisons in this specific situation. This is further explained by Levin et al., 1994 and discussed in Maxwell and Delaney, 2004 (Chapter 5) and Cohen, 2001 (Chapter 12).

*Fisher's LSD correction has been reported also for survival data but it doesn't apply to this kind of data. Also here we need a (Bonferroni) correction. For the survival data, power calculations were performed with the Sample Size Calculator, however I do not have a clear link to the software. The authors should provide all the used parameters and references*.

Thank you for this correction. We have updated the power calculations to reflect this adjustment. The link to the online calculator used is included as a hyperlink in the manuscript and should direct you to here: http://www.sample-size.net/sample-size-survival-analysis/, which includes the reference (28) for the formulas used.

Additionally, screen shots of the input and output parameters are included on the project page on the Open Science Framework.

*References*:

Levin, J.R., Serline, R.C., & Seaman M.A. (1994). A controlled, powerful multiple-comparison strategy for several situations. Psychological Bulletin, 115, 153–159.

Maxwell, S.E. & Delaney, H.D. (2004). Designing experiments and analyzing data: a model comparison perspecitive. Lawrence Erlbaum Associates, Mahwah, N.J., second edition.

Cohen, B.H. (2001). Explaining psychological statistics. John Wiley and Sons, New York, second edition.